# PLASTICINELAB: A SOFT-BODY MANIPULATION BENCHMARK WITH DIFFERENTIABLE PHYSICS

**Zhiao Huang** [*]
UC San Diego
z2huang@eng.ucsd.edu

**Yuanming Hu**
MIT
yuanming@mit.edu

**Tao Du**
MIT
taodu@csail.mit.edu

**Siyuan Zhou**
Peking University
elegyhunter@gmail.com

**Hao Su**
UC San Diego
haosu@eng.ucsd.edu

**Joshua B. Tenenbaum**
MIT BCS, CBMM, CSAIL
jbt@mit.edu

**Chuang Gan**
MIT-IBM Watson AI Lab
ganchuang@csail.mit.edu

## ABSTRACT

Simulated virtual environments serve as one of the main driving forces behind developing and evaluating skill learning algorithms. However, existing environments typically only simulate rigid body physics. Additionally, the simulation process usually does not provide gradients that might be useful for planning and control optimizations. We introduce a new differentiable physics benchmark called *PasticineLab*, which includes a diverse collection of soft body manipulation tasks. In each task, the agent uses manipulators to deform the plasticine into a desired configuration. The underlying physics engine supports *differentiable elastic and plastic deformation* using the DiffTaichi system, posing many underexplored challenges to robotic agents. We evaluate several existing reinforcement learning (RL) methods and gradient-based methods on this benchmark. Experimental results suggest that 1) RL-based approaches struggle to solve most of the tasks efficiently; 2) gradient-based approaches, by optimizing open-loop control sequences with the built-in differentiable physics engine, can rapidly find a solution within tens of iterations, but still fall short on multi-stage tasks that require long-term planning. We expect that PlasticineLab will encourage the development of novel algorithms that combine differentiable physics and RL for more complex physics-based skill learning tasks. PlasticineLab is publicly available [1].

## 1 INTRODUCTION

Virtual environments, such as Arcade Learning Environment (ALE) (Bellemare et al., 2013), MuJoCo (Todorov et al., 2012), and OpenAI Gym (Brockman et al., 2016), have significantly benefited the development and evaluation of learning algorithms on intelligent agent control and planning. However, existing virtual environments for skill learning typically involves *rigid-body* dynamics only. Research on establishing standard *soft-body* environments and benchmarks is sparse, despite the wide range of applications of soft bodies in multiple research fields, e.g., simulating virtual surgery in healthcare, modeling humanoid characters in computer graphics, developing biomimetic actuators in robotics, and analyzing fracture and tearing in material science.

Compared to its rigid-body counterpart, soft-body dynamics is much more intricate to simulate, control, and analyze. One of the biggest challenges comes from its infinite degrees of freedom (DoFs) and the corresponding high-dimensional governing equations. The intrinsic complexity of soft-body dynamics invalidates the direct application of many successful robotics algorithms designed for rigid

---

[*]This work was done during an internship at the MIT-IBM Watson AI Lab.
[1]Project page: http://plasticinelab.csail.mit.edu

bodies only and inhibits the development of a simulation benchmark for evaluating novel algorithms tackling soft-body tasks.

In this work, we aim to address this problem by proposing PlasticineLab, a novel benchmark for running and evaluating 10 soft-body manipulation tasks with 50 configurations in total. These tasks have to be performed by complex operations, including pinching, rolling, chopping, molding, and carving. Our benchmark is highlighted by the adoption of *differentiable physics* in the simulation environment, providing for the first time analytical gradient information in a soft-body benchmark, making it possible to conduct supervised learning with gradient-based optimization. In terms of the soft-body model, we choose to study plasticine (Fig. 1, left), a versatile elastoplastic material for sculpturing. Plasticine deforms elastically under small deformation, and plastically under large deformation. Compared to regular elastic soft bodies, plasticine establishes more diverse and realistic behaviors and brings challenges unexplored in previous research, making it a representative medium to test soft-body manipulation algorithms (Fig. 1, right).

We implement PlasticineLab, its gradient support, and its elastoplastic material model using Taichi (Hu et al., 2019a), whose CUDA backend leverages massive parallelism on GPUs to simulate a diverse collection of 3D soft-bodies in real time. We model the elastoplastic material using the Moving Least Squares Material Point Method (Hu et al., 2018) and the von Mises yield criterion. We use Taichi's two-scale reverse-mode differentiation system (Hu et al., 2020) to automatically compute gradients, including the numerically challenging SVD gradients brought by the plastic material model. With full gradients at hand, we evaluated gradient-based planning algorithms on all soft-robot manipulation tasks in PlasticineLab and compared its efficiency to RL-based methods. Our experiments revealed that gradient-based planning algorithms could find a more precious solution within tens of iterations with the extra knowledge of the physical model. At the same time, RL methods may fail even after 10K episodes. However, gradient-based methods lack enough momentum to resolve long-term planning, especially on multi-stage tasks. These findings have deepened our understanding of RL and gradient-based planning algorithms. Additionally, it suggests a promising direction of combining both families of methods' benefits to advance complex planning tasks involving soft-body dynamics. In summary, we contribute in this work the following:

- We introduce, to the best of our knowledge, the first skill learning benchmark involving elastic and plastic soft bodies.

- We develop a fully-featured differentiable physical engine, which supports elastic and plastic deformation, soft-rigid material interaction, and a tailored contact model for differentiability.

- The broad task coverage in the benchmark enables a systematic evaluation and analysis of representative RL and gradient-based planning algorithms. We hope such a benchmark can inspire future research to combine differentiable physics with imitation learning and RL.

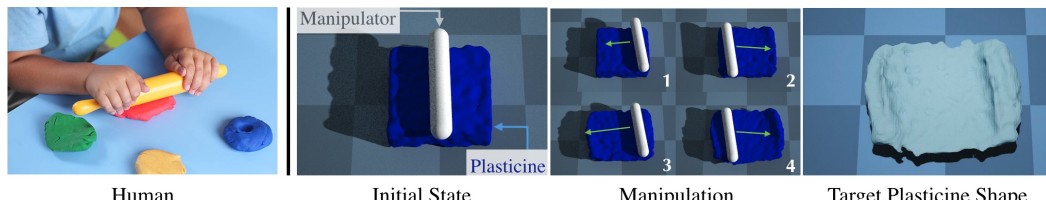

| Human | Initial State | Manipulation | Target Plasticine Shape |

Figure 1: **Left:** A child deforming a piece of plasticine into a thin pie using a rolling pin. **Right:** The challenging **RollingPin** scene in PlasticineLab. The agent needs to flatten the material by rolling the pin back and forth, so that the plasticine deforms into the target shape.

## 2 RELATED WORK

**Learning in virtual environments** Recently, several simulation platforms and datasets have been developed to facilitate the research and development of new algorithms on RL and robotics. An incomplete list includes RL Benchmark (Duan et al., 2016), DeepMind Lab (Beattie et al., 2016), OpenAI Gym (Brockman et al., 2016), AI2-THOR (Kolve et al., 2017), VirtualHome (Puig et al., 2018), Gibson (Xia et al., 2018), Habitat (Savva et al., 2019), SAPIEN (Xiang et al., 2020), and

TDW (Gan et al., 2020). We observe a tendency to use full-physics simulators with realistic dynamics. However, most of these virtual environments are based on rigid-body physical engines, such as MuJoCo (Todorov et al., 2012) and PyBullet (Coumans & Bai, 2016). While some support soft-body dynamics in theory (e.g., TDW and SAPIEN is based on NVIDIA PhysX (PhysX) that supports particle simulation), none has provided the assets and tasks for soft-body manipulation. Differentiable information is also missing in these engines. We fill in this gap with our PlasticineLab benchmark.

**Differentiable physics engines** Differentiable physics engines for machine learning have gained increasing popularity. One family of approaches *approximates* physical simulators using neural networks, which are naturally differentiable (Battaglia et al., 2016; Chang et al., 2016; Mrowca et al., 2018; Li et al., 2018). A more direct and accurate approach is to implement physics-based simulators using differentiable programming systems, e.g., standard deep learning frameworks equipped with automatic differentiation tools (Degrave et al., 2016; de Avila Belbute-Peres et al., 2018; Schenck & Fox, 2018; Heiden et al., 2019). These systems are typically redistricted to explicit time integration. Other approaches of evaluating simulation gradient computation include using the adjoint methods to differentiate implicit time integrators (Bern et al., 2019; Geilinger et al., 2020), LCP (de Avila Belbute-Peres et al., 2018) and leveraging QR decompositions (Liang et al., 2019; Qiao et al., 2020). Closely related to our work is ChainQueen (Hu et al., 2019b), a differentiable simulator for *elastic* bodies, and DiffTaichi (Hu et al., 2020), a system to automatically generate high-performance simulation gradient kernels. Our simulator is originated from ChainQueen but with significant modifications in order to add our novel support for *plasticity and contact* gradients.

**Trajectory optimization** Our usage of differentiable simulation in planning soft-body manipulation is closely related to trajectory optimization, a topic that has been extensively studied in robotics for years and has been applied to terrestrial robots (Posa et al., 2014; Erez & Todorov, 2012; de Avila Belbute-Peres et al., 2018), aerial robots (Foehn et al., 2017; Tang & Kumar, 2015; Sreenath et al., 2013), and, closest to examples in our work, robotic manipulators (Marchese et al., 2016; Li et al., 2015). Both trajectory optimization and differentiable physics formulate planning as an optimization problem and derive gradients from governing equations of the dynamics (Tedrake, 2020). Still, the problem of motion planning for soft-body manipulation is under exploration in both communities because of two challenges: first, the high DoFs in soft-body dynamics make traditional trajectory optimization methods computationally prohibitive. Second, and more importantly, contacts between soft bodies are intricate to formulate in a concise manner. Our differentiable physics simulator addresses both issues with the recent development of DiffTaichi (Hu et al., 2020), unlocking gradient-based optimization techniques on planning for soft-body manipulation with high DoFs ($> 10,000$) and complex contact.

**Learning-based soft body manipulation** Finally, our work is also relevant to prior methods that propose learning-based techniques for manipulating physics systems with high degrees of freedom, e.g. cloth (Liang et al., 2019; Wu et al., 2020), fluids (Ma et al., 2018; Holl et al., 2020), and rope (Yan et al., 2020; Wu et al., 2020). Compared to our work, all of these prior papers focused on providing solutions to specific robot instances, while the goal of our work is to propose a comprehensive benchmark for evaluating and developing novel algorithms in soft-body research. There are also considerable works on soft manipulators (George Thuruthel et al., 2018; Della Santina et al., 2018). Different from them, we study soft body manipulation with rigid manipulators.

# 3 THE PLASTICINELAB LEARNING ENVIRONMENT

PlasticineLab is a collection of challenging soft-body manipulation tasks powered by a differentiable physics simulator. All tasks in PlasticineLab require an agent to deform one or more pieces of 3D plasticine with rigid-body manipulators. The underlying simulator in PlasticineLab allows users to execute complex operations on soft bodies, including pinching, rolling, chopping, molding, and carving. We introduce the high-level design of the learning environment in this section and leave the technical details of the underlying differentiable simulator in Sec. 4.

## 3.1 TASK REPRESENTATION

PlasticineLab presents 10 tasks with the focus on soft-body manipulation. Each task contains one or more soft bodies and a kinematic manipulator, and the end goal is to deform the soft body into a target shape with the planned motion of the manipulator. Following the standard reinforcement

learning framework (Brockman et al., 2016), the agent is modeled with the Markov Decision Process (MDP), and the design of each task is defined by its state and observation, its action representation, its goal definition, and its reward function.

**Markov Decision Process** An MDP contains a state space $\mathcal{S}$, an action space $\mathcal{A}$, a reward function $\mathcal{R} : \mathcal{S} \times \mathcal{A} \times \mathcal{S} \rightarrow \mathbb{R}$, and a transition function $\mathcal{T} : \mathcal{S} \times \mathcal{A} \rightarrow \mathcal{S}$. In PlasticineLab, the physics simulator determines the transition between states. The goal of the agent is to find a stochastic policy $\pi(a|s)$ to sample action $a \in \mathcal{A}$ given state $s \in \mathcal{S}$, that maximizes the expected cumulative future return $E_\pi \left[ \sum_{t=0}^{\infty} \gamma^t \mathcal{R}(s_t, a_t) \right]$ where $0 < \gamma < 1$ is the discount factor.

**State** The state of a task includes a proper representation of soft bodies and the end effector of the kinematic manipulator. Following the widely used particle-based simulation methodology in previous work, we represent soft-body objects as a particle system whose state includes its particles' positions, velocities, and strain and stress information. Specifically, the particle state is encoded as a matrix of size $N_p \times d_p$ where $N_p$ is the number of particles. Each row in the matrix consists of information from a single particle: two 3D vectors for position and velocities and two 3D matrices for deformation gradients and affine velocity fields (Jiang et al., 2015), all of which are stacked together and flattened into a $d_p$-dimensional vector.

Being a kinematic rigid body, the manipulator's end effector is compactly represented by a 7D vector consisting of its 3D position and orientation represented by a 4D quaternion, although some DoFs may be disabled in certain scenes. For each task, this representation results in an $N_m \times d_m$ matrix encoding the full states of manipulators, where $N_m$ is the number of manipulators needed in the task and $d_m = 3$ or $7$ depending on whether rotation is needed. Regarding the interaction between soft bodies and manipulators, we implement one-way coupling between rigid objects and soft bodies and fix all other physical parameters such as particle's mass and manipulator's friction.

**Observation** While the particle states fully characterize the soft-body dynamics, its high DoFs are hardly tractable for any planning and control algorithm to work with directly. We thus downsample $N_k$ particles as landmarks and stack their positions and velocities (6D for each landmark) into a matrix of size $N_k \times 6$, which serves as the observation of the particle system. Note that landmarks in the same task have fixed relative locations in the plasticine's initial configuration, leading to a consistent particle observation across different configurations of the task. Combining the particle observation with the manipulator state, we end up having $N_k \times 6 + N_m \times d_m$ elements in the observation vector.

**Action** At each time step, the agent is instructed to update the linear (and angular when necessary) velocities of the manipulators in a kinematic manner, resulting in an action of size $N_m \times d_a$ where $d_a = 3$ or $6$ depending on whether rotations of the manipulators are enabled in the task. For each task, we provide global $A_{\min}, A_{\max} \in \mathbb{R}^{d_a}$, the lower and upper bounds on the action, to stabilize the physics simulation.

**Goal and Reward** Each task is equipped with a target shape represented by its mass tensor, which is essentially its density field discretized into a regular grid of size $N_{grid}^3$. At each time step $t$, we compute the mass tensor of the current soft body $S_t$. Discretizing both target and current shapes into a grid representation allows us to define their similarity by comparing densities at the same locations, avoiding the challenging problem of matching particle systems or point clouds. The complete definition of our reward function includes a similarity metric as well as two regularizers on the high-level motion of the manipulator: $\mathcal{R} = -c_1 \mathcal{R}_{\mathrm{mass}} - c_2 \mathcal{R}_{\mathrm{dist}} - c_3 \mathcal{R}_{\mathrm{grasp}} + c_4$, where $\mathcal{R}_{\mathrm{mass}}$ measures the $L_1$ distance between the two shapes' mass tensors as described above, $\mathcal{R}_{\mathrm{dist}}$ is the dot product of the signed distance field (SDF) of the target shape and the current shape's mass tensor, and $\mathcal{R}_{\mathrm{grasp}}$ encourages the manipulators to be closer to the soft bodies. Positive weights $c_1, c_2, c_3$ are constant for all tasks. The bias $c_4$ is selected for each environment to ensure the reward is nonnegative initially.

## 3.2 Evaluation Suite

PlasticineLab has a diverse collection of 10 tasks (Fig. 2). We describe four representative tasks here, and the remaining six tasks are detailed in Appendix B.

These tasks, along with their variants in different configurations, form an evaluation suite for benchmarking performance of soft-body manipulation algorithms. Each task has 5 variants (50 config-

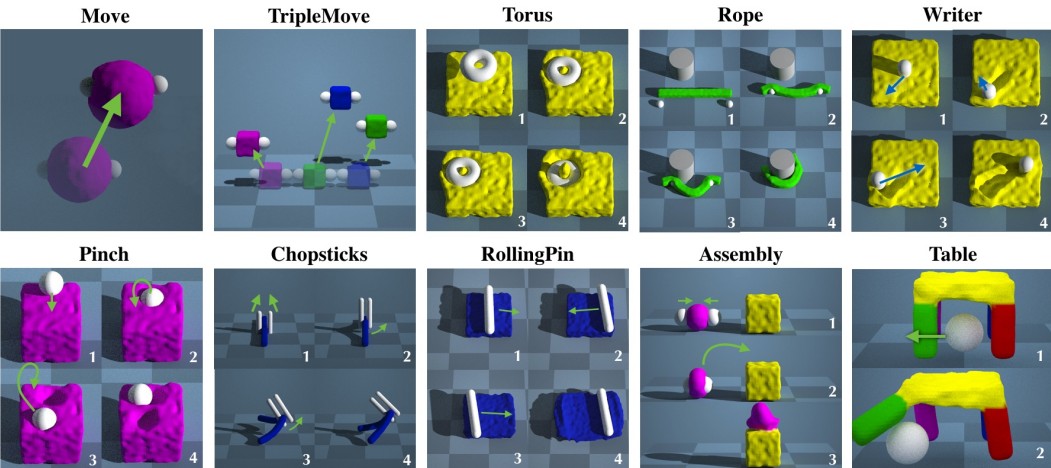

Figure 2: Tasks and reference solutions of PlasticineLab. Certain tasks require multi-stage planning.

urations in total) generated by perturbing the initial and target shapes and the initial locations of manipulators.

**Rope** The agent needs to wind a rope, modeled as a long plasticine piece, around a rigid pillar with two spherical manipulators. The pillar's position varies in different configurations.

**Writer** The agent manipulates a "pen" (represented using a vertical capsule), to sculpt a target scribble on cubic plasticine. For each configuration, we generate the scribble by drawing random 2D lines on the plasticine surface. The three-dimensional action controls the tip of the pen.

**Chopsticks** The agent uses a pair of chopsticks, modeled as two parallel capsules, to pick up the rope on the ground and rotate it into the target location. The manipulator has 7 DoFs: 6 for moving and rotating the pair of chopsticks and 1 for controlling the distance between them.

**RollingPin** The agent learns to flatten a "pizza dough", which is modeled as a plasticine box, with a rigid rolling pin. We simulate the rolling pin with a 3-DoF capsule: 1) the pin can descend vertically to press the dough; 2) the pin can rotate along the vertical axis to change its orientation; 3) the agent can also roll the pin over the plasticine to flatten it.

## 4 DIFFERENTIABLE ELASTOPLASTICITY SIMULATION

The simulator is implemented using Taichi (Hu et al., 2019a) and runs on CUDA. Continuum mechanics is discretized using the Moving Least Squares Material Point Method (MLS-MPM) (Hu et al., 2018), a simpler and more efficient variant of the B-spline Material Point Method (MPM) in computer graphics (Stomakhin et al., 2013). Both Lagrangian particles and Eulerian background grids are used in the simulator. Material properties, include position, velocity, mass, density, and deformation gradients, are stored on Lagrangian particles that move along with the material, while particle interactions and collisions with rigid bodies are handled on the background Eulerian grid. We refer the reader to ChainQueen (Hu et al., 2019b) and DiffTaichi (Hu et al., 2020) for more details on differentiable MPM with elastic materials. Here we focus on extending the material model with *(differentiable) plasticity*, a defining feature of plasticine. We leverage Taichi's reverse-mode automatic differentiation system (Hu et al., 2020) for most of the gradient evaluations.

**von Mises yield criterion** We use a simple von Mises yield criterion for modeling plasticity, following the work of Gao et al. (2017). According to the von Mises yield criterion, a plasticine particle yields (i.e., deforms plastically) when its second invariant of the deviatoric stress exceeds a certain threshold, and a projection on the deformation gradient is needed since the material "forgets" its rest state. This process is typically called *return mapping* in MPM literature.

**Return mapping and its gradients** Following Klár et al. (2016) and Gao et al. (2017), we implement the return mapping as a 3D projection process on the singular values of the deformation gra-

dients of each particle. This means we need a singular value decomposition (SVD) process on the particles' deformation gradients, and we provide the pseudocode of this process in Appendix A. For backpropagation, we need to evaluate *gradients of SVD*. Taichi's internal SVD algorithm (McAdams et al., 2011) is iterative, which is numerically unstable when automatically differentiated in a brute-force manner. We use the approach in Townsend (2016) to differentiate the SVD. For zeros appearing in the denominator when singular values are not distinct, we follow Jiang et al. (2016) to push the absolute value of the denominators to be greater than $10^{-6}$.

**Contact model and its softened version for differentiability** We follow standard MPM practices and use grid-base contact treatment with Coulomb friction (see, for example, Stomakhin et al. (2013)) to handle soft body collision with the floor and the rigid body obstacles/manipulators. Rigid bodies are represented as time-varying SDFs. In classical MPM, contact treatments induce a drastic non-smooth change of velocities along the rigid-soft interface. To improve reward smoothness and gradient quality, we use a *softened* contact model during backpropagation. For any grid point, the simulator computes its signed distance $d$ to the rigid bodies. We then compute a smooth *collision strength factor* $s = \min\{\exp(-\alpha d), 1\}$, which increases exponentially when $d$ decays until $d = 0$. Intuitively, collision effects get stronger when rigid bodies get closer to the grid point. The positive parameter $\alpha$ determines the sharpness of the softened contact model. We linearly blend the grid point velocity before and after collision projection using factor $s$, leading to a smooth transition zone around the boundary and improved contact gradients.

## 5 EXPERIMENTS

### 5.1 EVALUATION METRICS

We first generate five configurations for each task, resulting in 50 different reinforcement learning configurations. We compute the normalized incremental IoU score to measure if the state reaches the goal. We apply the soft IoU (Rahman & Wang, 2016) to estimate the distance between a state and the goal. We first extract the grid mass tensor $S$, the masses on all grids. Each value $S^{xyz}$ stores how many materials are located in the grid point $(x, y, z)$, which is always nonnegative. Let two states' 3D mass tensors be $S_1$ and $S_2$. We first divide each tensor by their maximum magnitude to normalize its values to $[0, 1]$: $\bar{S}_1 = \frac{S_1}{\max_{ijk} S_1^{ijk}}$ and $\bar{S}_2 = \frac{S_2}{\max_{ijk} S_2^{ijk}}$. Then the softened IoU of the two state is calculated as $IoU(S_1, S_2) = \frac{\sum_{ijk} \bar{S}_1 \bar{S}_2}{\sum_{ijk} \bar{S}_1 + \bar{S}_2 - \bar{S}_1 \bar{S}_2}$. We refer readers to Appendix F for a better explanation for the soft IoU. The final normalized incremental IoU score measures how much IoU increases at the end of the episode than the initial state. For the initial state $S_0$, the last state $S_t$ at the end of the episode, and the goal state $S_g$, the normalized incremental IoU score is defined as $\frac{IoU(S_t, S_g) - IoU(S_0, S_g)}{1 - IoU(S_0, S_g)}$. For each task, we evaluate the algorithms on five configurations and report an algebraic average score.

### 5.2 EVALUATIONS ON REINFORCEMENT LEARNING

We evaluate the performance of the existing RL algorithms on our tasks. We use three SOTA model-free reinforcement learning algorithms: Soft Actor-Critic (SAC) (Haarnoja et al., 2017), Twin Delayed DDPG (TD3) (Fujimoto et al., 2018), and Policy Proximal Optimization (PPO) (Schulman et al., 2017). We train each algorithm on each configuration for 10000 episodes, with 50 environment steps per episode.

Figure 3 shows the normalized incremental IoU scores of the tested reinforcement learning algorithms on each scene. Most RL algorithms can learn reasonable policies for **Move**. However, RL algorithms can hardly match the goal shape exactly, which causes a small defect in the final shape matching. We notice that it is common for the RL agent to release the objects during exploration, leading to a free-fall of plasticine under gravity. Then it becomes challenging for the agent to regrasp the plasticine, leading to training instability and produces unsatisfactory results. The same in **Rope**, agents can push the rope towards the pillar and gain partial rewards, but they fail to move the rope around the pillar in the end. Increasing the numbers of manipulators and plasticine boxes causes significant difficulties in **TripleMove** for RL algorithms, revealing their deficiency in scaling to high dimensional tasks. In **Torus**, the performance seems to depend on the initial policy. They could

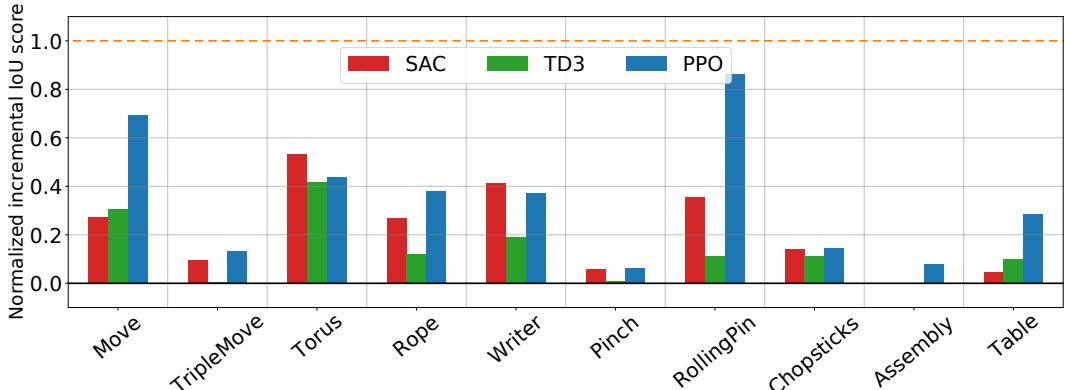

Figure 3: The final normalized incremental IoU score achieved by RL methods within $10^4$ epochs. Scores lower than 0 are clamped. The dashed orange line indicates the theoretical upper limit.

sometimes find a proper direction to press manipulators, but occasionally, they fail as manipulators never touch the plasticine, generating significant final score variance. Generally, we find that PPO performs better than the other two. In **RollingPin**, both SAC and PPO agents find the policy to go back and forth to flatten the dough, but PPO generates a more accurate shape, resulting in a higher normalized incremental IoU score. We speculate that our environment favors PPO over algorithms dependent on MLP critic networks. We suspect it is because PPO benefits from on-policy samples while MPL critic networks might not capture the detailed shape variations well.

In some harder tasks, like **Chopsticks** that requires the agent to carefully handle the 3D rotation, and **Writer** that requires the agent to plan for complex trajectories for carving the traces, the tested algorithm seldom finds a reasonable solution within the limited time ($10^4$ episodes). In **Assembly**, all agents are stuck in local minima easily. They usually move the spherical plasticine closer to the destination but fail to lift it up to achieve an ideal IoU. We expect that a carefully designed reward shaping, better network architectures, and fine-grained parameter tuning might be beneficial in our environments. In summary, plasticity, together with the soft bodies' high DoFs, poses new challenges for RL algorithms.

## 5.3 EVALUATIONS ON DIFFERENTIABLE PHYSICS FOR TRAJECTORY OPTIMIZATION

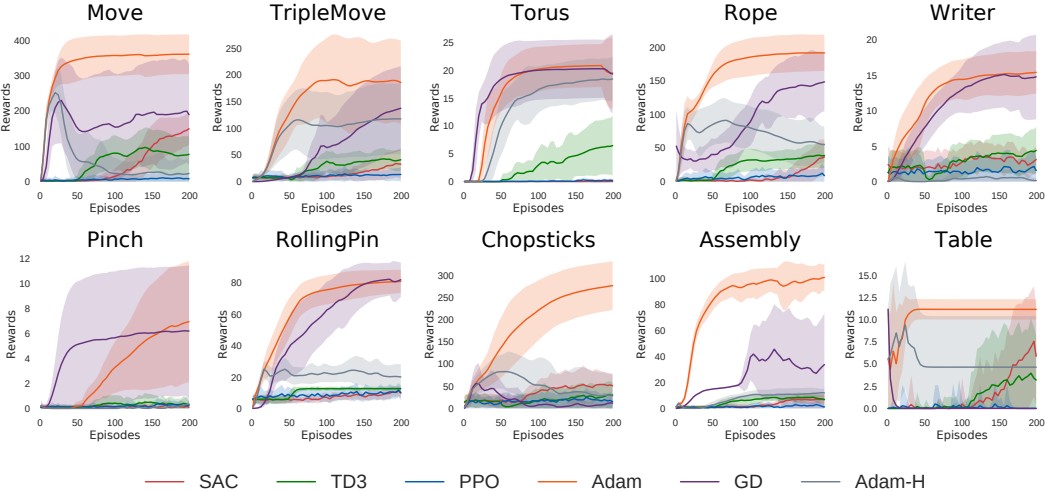

Figure 4: Rewards and their variances in each task w.r.t. the number of episodes spent on training. We clamp the reward to be greater than 0 for a better illustration.

Thanks to the built-in differentiable physics engine in PlasticineLab, we can apply gradient-based optimization to plan open-loop action sequences for our tasks. In gradient-based optimization, for a

| **Env** | Move | Tri. Move | Torus | Rope | Writer |
|---|---|---|---|---|---|
| SAC | $0.27 \pm 0.27$ | $0.12 \pm 0.11$ | $\mathbf{0.53 \pm 0.42}$ | $0.29 \pm 0.20$ | $\mathbf{0.41 \pm 0.23}$ |
| TD3 | $0.31 \pm 0.20$ | $0.00 \pm 0.02$ | $0.42 \pm 0.42$ | $0.12 \pm 0.15$ | $0.19 \pm 0.18$ |
| PPO | $\mathbf{0.69 \pm 0.15}$ | $\mathbf{0.13 \pm 0.09}$ | $0.44 \pm 0.38$ | $\mathbf{0.38 \pm 0.19}$ | $0.37 \pm 0.17$ |
| Adam | $\mathbf{0.90 \pm 0.12}$ | $\mathbf{0.35 \pm 0.20}$ | $0.77 \pm 0.39$ | $\mathbf{0.59 \pm 0.13}$ | $0.62 \pm 0.27$ |
| GD | $0.51 \pm 0.39$ | $0.24 \pm 0.17$ | $\mathbf{0.79 \pm 0.37}$ | $0.41 \pm 0.17$ | $\mathbf{0.69 \pm 0.29}$ |
| Adam-H | $0.05 \pm 0.15$ | $0.26 \pm 0.15$ | $0.72 \pm 0.23$ | $0.21 \pm 0.09$ | $0.00 \pm 0.00$ |

| **Env** | Pinch | RollingPin | Chopsticks | Assembly | Table |
|---|---|---|---|---|---|
| SAC | $0.05 \pm 0.08$ | $0.36 \pm 0.30$ | $0.13 \pm 0.08$ | $0.00 \pm 0.00$ | $0.04 \pm 0.12$ |
| TD3 | $0.01 \pm 0.02$ | $0.11 \pm 0.02$ | $0.11 \pm 0.07$ | $0.00 \pm 0.00$ | $0.10 \pm 0.16$ |
| PPO | $\mathbf{0.06 \pm 0.09}$ | $\mathbf{0.86 \pm 0.10}$ | $\mathbf{0.14 \pm 0.09}$ | $0.06 \pm 0.17$ | $\mathbf{0.29 \pm 0.28}$ |
| Adam | $\mathbf{0.08 \pm 0.08}$ | $\mathbf{0.93 \pm 0.04}$ | $\mathbf{0.88 \pm 0.08}$ | $\mathbf{0.90 \pm 0.10}$ | $0.01 \pm 0.01$ |
| GD | $0.03 \pm 0.05$ | $0.89 \pm 0.11$ | $0.03 \pm 0.04$ | $0.27 \pm 0.36$ | $0.00 \pm 0.00$ |
| Adam-H | $0.00 \pm 0.02$ | $0.26 \pm 0.12$ | $0.02 \pm 0.06$ | $0.03 \pm 0.03$ | $\mathbf{0.00 \pm 0.01}$ |

Table 1: The averaged normalized incremental IoU scores and the standard deviations of each method. Adam-H stands for optimizing on the hard contact model with Adam optimizer. We train RL agent for 10000 episodes and optimizing for 200 episodes for gradient-based approaches.

certain configuration starting at state $s$, we initialize a random action sequence $\{a_1, \ldots, a_T\}$. The simulator will simulate the whole trajectory, accumulate the reward at each time step, and do back-propagation to compute the gradients of all actions. We then apply a gradient-based optimization method to maximize the sum of rewards. We assume all information of the environment is known. This approach's goal is not to find a controller that can be executed in the real world. Instead, we hope that differentiable physics can help find a solution efficiently and pave roads for other control or reinforcement/imitation learning algorithms.

In Figure 4, we demonstrate the optimization efficiency of differentiable physics by plotting the reward curve w.r.t. the number of environment episodes and compare different variants of gradient descent. We test the Adam optimizer (Adam) and gradient descent with momentum (GD). We use the soft contact model to compute the gradients. We compare the Adam optimizer with a hard contact model (Adam-H). For each optimizer, we modestly choose a learning rate of $0.1$ or $0.01$ per task to handle the different reward scales across tasks. Notice that we only use the soft contact model for computing the gradients and search for a solution. We evaluate all solutions in environments with hard contacts. In Figure 4, we additionally plot the training curve of reinforcement learning algorithms to demonstrate the efficiency of gradient-based optimization. Results show that optimization-based methods can find a solution for challenging tasks within tens of iterations. Adam outperforms GD in most tasks. This may be attributed to Adam's adaptive learning rate scaling property, which fits better for the complex loss surface of the high-dimensional physical process. The hard contact model (Adam-H) performs worse than the soft version (Adam) in most tasks, which validates the intuition that a soft model is generally easier to optimize.

Table 1 lists the normalized incremental IoU scores, together with the standard deviations of all approaches. The full knowledge of the model provides differentiable physics a chance to achieve more precious results. Gradient descent with Adam can find the way to move the rope around the pillar in **Rope**, jump over the sub-optimal solution in **Assembly** to put the sphere above the box, and use the chopsticks to pick up the rope. Even for **Move**, it often achieves better performance by better aligning with the target shape and a more stable optimization process.

Some tasks are still challenging for gradient-based approaches. In **TripleMove**, the optimizer minimizes the particles' distance to the closet target shape, which usually causes two or three plasticines to crowd together into one of the target locations. It is not easy for the gradient-based approaches, which have no exploration, to jump out such local minima. The optimizer also fails on the tasks that require multistage policies, e.g., **Pinch** and **Writer**. In **Pinch**, the manipulator needs to press the objects, release them, and press again. However, after the first touch of the manipulator and the plasticine, any local perturbation of the spherical manipulator doesn't increase the reward immediately, and the optimizer idles at the end. We also notice that gradient-based methods are sensitive to initialization. Our experiments initialize the action sequences around 0, which gives a good performance in most environments.

# 6    POTENTIAL RESEARCH PROBLEMS TO STUDY USING PLASTICINELAB

Our environment opens ample research opportunities for learning-based soft-body manipulation. Our experiments show that differential physics allows gradient-based trajectory optimization to solve simple planning tasks extremely fast, because gradients provide strong and clear guidance to improve the policy. However, gradients will vanish if the tasks involve detachment and reattachment between the manipulators and the plasticine. When we fail to use gradient-based optimization that is based on local perturbation analysis, we may consider those methods that allow multi-step exploration and collect cumulative rewards, e.g., random search and reinforcement learning. Therefore, it is interesting to study how differentiable physics may be combined with these sampling-based methods to solve planning problems for soft-body manipulation.

Beyond the planning problem, it is also interesting to study how we shall design and learn effective controllers for soft-body manipulation in this environment. Experimental results (Sec. 5.2) indicate that there is adequate room for improved controller design and optimization. Possible directions include designing better reward functions for RL and investigating proper 3D deep neural network structures to capture soft-body dynamics.

A third interesting direction is to transfer the trained policy in PlasticineLab to the real world. While this problem is largely unexplored, we believe our simulator can help in various ways: 1. As shown in Gaume et al. (2018), MPM simulation results can accurately match the real world. In the future, we may use our simulator to plan a high-level trajectory for complex tasks and then combine with low-level controllers to execute the plan. 2. Our differential simulator can compute the gradient to physical parameters and optimize parameters to fit the data, which might help to close the sim2real gap. 3. PlasticineLab can also combine domain randomization and other sim2real methods (Matas et al., 2018). One can customize physical parameters and the image renderer to implement domain randomization in our simulator. We hope our simulator can serve as a good tool to study real-world soft-body manipulation problems.

Finally, generalization is an important exploration direction. Our platform supports procedure generation and can generate and simulate various configurations with different objects, evaluating different algorithms' generalizability. PlasticineLab is a good platform to design rich goal-condition tasks, and we hope it can inspire future work.

# 7    CONCLUSION AND FUTURE WORK

We presented PlasticineLab, a new differentiable physics benchmark for soft-body manipulation. To the best of our knowledge, PlasticineLab is the first skill-learning environment that simulates elastoplastic materials while being differentiable. The rich task coverage of PlasticineLab allows us to systematically study the behaviors of state-of-the-art RL and gradient-based algorithms, providing clues to future work that combines the two families of methods.

We also plan to extend the benchmark with more articulation systems, such as virtual shadow hands[2]. As a principled simulation method that originated from the computational physics community (Sulsky et al., 1995), MPM is convergent under refinement and has its own accuracy advantages. However, modeling errors are inevitable in virtual environments. Fortunately, apart from serving as a strong supervision signal for planning, the simulation gradient information can also guide systematic identification. This may allow robotics researchers to "optimize" tasks themselves, potentially simultaneously with controller optimization, so that sim-to-real gaps are automatically minimized.

We believe PlasticineLab can significantly lower the barrier of future research on soft-body manipulation skill learning, and will make its unique contributions to the machine learning community.

**Acknowledgement** This work is in part supported by ONR MURI N00014-16-1-2007, the Center for Brain, Minds, and Machines (CBMM, funded by NSF STC award CCF-1231216), and IBM Research.

---

[2]https://en.wikipedia.org/wiki/Shadow_Hand

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

## A  SIMULATOR IMPLEMENTATION DETAILS

**von Mises plasticity return mapping pseudo code**   Here we list the implementation of the forward return mapping (Gao et al., 2017). Note the SVD in the beginning leads to gradient issues that need special treatments during backpropagation.

```
def von_Mises_return_mapping(F):
    # F is the deformation gradient before return mapping

    U, sig, V = ti.svd(F)

    epsilon = ti.Vector([ti.log(sig[0, 0]), ti.log(sig[1, 1])])
    epsilon_hat = epsilon - (epsilon.sum() / 2)
    epsilon_hat_norm = epsilon_hat.norm()
    delta_gamma = epsilon_hat_norm - yield_stress / (2 * mu)

    if delta_gamma > 0: # Yields!
        epsilon -= (delta_gamma / epsilon_hat_norm) * epsilon_hat
        sig = make_matrix_from_diag(ti.exp(epsilon))
        F = U @ sig @ V.transpose()
    return F
```

**Parameters**   We use a yield stress of 50 for plasticine in all tasks except **Rope**, where we use 200 as the yield stress to prevent the rope from fracturing. We use $\alpha = 666.7$ in the soft contact model.

**Parallelism and performance**   Our parallel mechanism is based on ChainQueen (Hu et al.,2019b). A single MPM substep involves three stages: particle to grid transform (p2g), grid boundary conditions (gridop), and grid to particle transform (g2p). In each stage, we use a parallel for-loop to loop over all particles (p2g) or grids (for gridop and g2p) to do physical computations and ap-ply atomic operators to write results back to particles/grids. Gradient computation, which needs to reverse the three stages, is automatically done by DiffTaich (Hu et al., 2020).

We benchmark our simulator's performance on each scene in Table 2. Note one step of our environment has 19 MPM substeps.

| **Env**     | Forward            | Forward + Backward |
|-------------|--------------------|--------------------|
| Move        | 14.26 ms (70 FPS)  | 35.62 ms (28 FPS)  |
| Tri.Move    | 17.81 ms (56 FPS)  | 41.88 ms (24 FPS)  |
| Torus       | 13.77 ms (73 FPS)  | 35.69 ms (28 FPS)  |
| Rope        | 15.05 ms (66 FPS)  | 38.70 ms (26 FPS)  |
| Writer      | 14.00 ms (71 FPS)  | 36.04 ms (28 FPS)  |
| Pinch       | 12.07 ms (83 FPS)  | 27.03 ms (37 FPS)  |
| RollingPin  | 14.14 ms (71 FPS)  | 36.45 ms (27 FPS)  |
| Chopsticks  | 14.24 ms (70 FPS)  | 35.68 ms (28 FPS)  |
| Assembly    | 14.43 ms (69 FPS)  | 36.51 ms (27 FPS)  |
| Table       | 14.00 ms (71 FPS)  | 35.49 ms (28 FPS)  |

Table 2: Performance on an NVIDIA GTX 1080 Ti GPU. We show the average running time for a single forward or forward + backpropagation step for each scene.

## B  MORE DETAILS ON THE EVALUATION SUITE

**Move** The agent uses two spherical manipulators to grasp the plasticine and move it to the target location. Each manipulator has 3 DoFs controlling its position only, resulting in a 6D action space.

**TripleMove** The agent operates three pairs of spherical grippers to relocate three plasticine boxes into the target positions. The action space has a dimension of 18. This task is challenging to both RL and gradient-based methods.

**Torus** A piece of cubic plasticine is fixed on the ground. In each configuration of the task, we generate the target shape by randomly relocating the plasticine and push a torus mold towards it. The agent needs to figure out the correct location to push down the mold.

**Pinch** In this task, the agent manipulates a rigid sphere to create dents on the plasticine box. The target shape is generated by colliding the sphere into the plasticine from random angles. To solve this task, the agent needs to discover the random motion of the sphere.

**Assembly** A spherical piece of plasticine is placed on the ground. The agent first deforms the sphere into a target shape and then moves it onto a block of plasticine. The manipulators are two spheres.

**Table** This task comes with a plasticine table with four legs. The agent pushes one of the table legs towards a target position using a spherical manipulator.

## C    REINFORCEMENT LEARNING SETUP

We use the open-source implementation of SAC, PPO and TD3 in our environments. We list part of the hyperparameters in Table 3 for SAC, Table 5 for TD3 and Table 4 for PPO. We fix $c_1 = 10, c_2 = 10$ and $c_3 = 1$ for all environments' reward.

Table 3: SAC Parameters

| | |
|---|---|
| gamma | 0.99 |
| policy lr | 0.0003 |
| entropy lr | 0.0003 |
| target update coef | 0.0003 |
| batch size | 256 |
| memory size | 1000000 |
| start steps | 1000 |

Table 4: PPO Parameters

| | |
|---|---|
| update steps | 2048 |
| lr | 0.0003 |
| entropy coef | 0 |
| value loss coef | 0.5 |
| batch size | 32 |
| horizon | 2500 |

Table 5: TD3 Parameters

| | |
|---|---|
| start timesteps | 1000 |
| batch size | 256 |
| gamma | 0.99 |
| tau | 0.005 |
| policy noise | 0.2 |
| noise clip | 0.5 |

## D    ABLATION STUDY ON YIELD STRESS

To study the effects of yield stress, we run experiments on a simple **Move** configuration (where SAC can solve it well) with different yield stress. We vary the yield stress from 10 to 1000 to generate 6 environments and train SAC on them. Figure 5 plots the agents' performances w.r.t. the number of training episodes. The agents achieve higher reward as the yield stress increase, especially in the beginning. Agents in high yield stress environments learn faster than those in lower yield stress environments. We attribute this to the smaller plastic deformation in higher yield stress environments. If we train the agents for more iterations, those in environments with yield stress larger than 100 usually converge to a same performance level, close to solving the problem. However, materials in environments with yield stress smaller than 100 tend to deform plastically, making it hard to grasp and move an object while not destroying its structure. This demonstrates a correlation between yield stress and task difficulty.

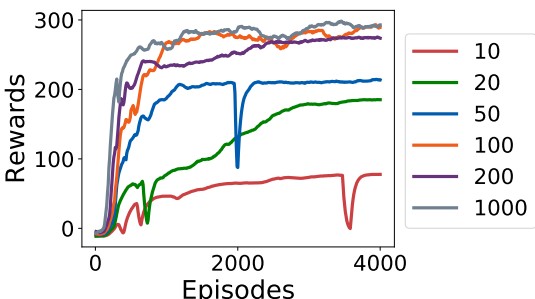

Figure 5: Rewards w.r.t. the number of training episode on 6 environments. Their yield stresses are 10, 20, 50, 100, 200, and 1000.

## E    DETAILED RESULTS ON ALL CONFIGURATIONS

We run each algorithm with 3 random seeds and each scene has 5 configurations. Therefore, $3 \times 5 = 15$ random seeds in total are used for each algorithm on each scene. We report the normalized incremental IoU scores and the standard deviations for each configuration in Table 6. We also show the optimization efficiency of differentiable physics in Figure 6.

| Method
Env | SAC | TD3 | PPO | Adam | GD | Adam-H |
|---|---|---|---|---|---|---|
| Move-v1 | $0.19 \pm 0.26$ | $0.30 \pm 0.11$ | $\mathbf{0.72 \pm 0.11}$ | $\mathbf{0.97 \pm 0.01}$ | $0.04 \pm 0.08$ | $0.00 \pm 0.00$ |
| Move-v2 | $0.20 \pm 0.28$ | $0.38 \pm 0.27$ | $\mathbf{0.60 \pm 0.10}$ | $0.67 \pm 0.03$ | $\mathbf{0.67 \pm 0.02}$ | $0.00 \pm 0.01$ |
| Move-v3 | $0.48 \pm 0.35$ | $0.29 \pm 0.27$ | $\mathbf{0.65 \pm 0.21}$ | $\mathbf{0.97 \pm 0.01}$ | $0.21 \pm 0.32$ | $0.00 \pm 0.00$ |
| Move-v4 | $0.19 \pm 0.17$ | $0.33 \pm 0.07$ | $\mathbf{0.79 \pm 0.07}$ | $\mathbf{0.99 \pm 0.01}$ | $0.73 \pm 0.30$ | $0.02 \pm 0.04$ |
| Move-v5 | $0.29 \pm 0.06$ | $0.23 \pm 0.16$ | $\mathbf{0.71 \pm 0.13}$ | $\mathbf{0.92 \pm 0.02}$ | $0.90 \pm 0.06$ | $0.22 \pm 0.28$ |
| Tri.Move-v1 | $\mathbf{0.10 \pm 0.09}$ | $0.00 \pm 0.00$ | $0.09 \pm 0.07$ | $\mathbf{0.26 \pm 0.22}$ | $0.05 \pm 0.08$ | $0.17 \pm 0.12$ |
| Tri.Move-v2 | $0.09 \pm 0.09$ | $0.00 \pm 0.00$ | $\mathbf{0.10 \pm 0.05}$ | $0.18 \pm 0.09$ | $\mathbf{0.25 \pm 0.00}$ | $0.24 \pm 0.02$ |
| Tri.Move-v3 | $0.04 \pm 0.06$ | $0.00 \pm 0.00$ | $\mathbf{0.11 \pm 0.05}$ | $\mathbf{0.23 \pm 0.01}$ | $0.15 \pm 0.08$ | $0.15 \pm 0.05$ |
| Tri.Move-v4 | $0.12 \pm 0.09$ | $0.00 \pm 0.00$ | $\mathbf{0.13 \pm 0.07}$ | $\mathbf{0.45 \pm 0.11}$ | $0.27 \pm 0.00$ | $0.23 \pm 0.03$ |
| Tri.Move-v5 | $0.23 \pm 0.09$ | $0.02 \pm 0.03$ | $\mathbf{0.24 \pm 0.09}$ | $\mathbf{0.62 \pm 0.01}$ | $0.49 \pm 0.15$ | $0.52 \pm 0.02$ |
| Torus-v1 | $0.48 \pm 0.28$ | $\mathbf{0.58 \pm 0.09}$ | $0.36 \pm 0.37$ | $0.90 \pm 0.00$ | $\mathbf{0.91 \pm 0.01}$ | $0.88 \pm 0.03$ |
| Torus-v2 | $\mathbf{0.37 \pm 0.35}$ | $0.00 \pm 0.00$ | $0.08 \pm 0.04$ | $0.30 \pm 0.43$ | $0.07 \pm 0.05$ | $\mathbf{0.39 \pm 0.10}$ |
| Torus-v3 | $0.40 \pm 0.49$ | $\mathbf{0.67 \pm 0.47}$ | $0.65 \pm 0.20$ | $\mathbf{1.00 \pm 0.00}$ | $\mathbf{1.00 \pm 0.00}$ | $\mathbf{1.00 \pm 0.00}$ |
| Torus-v4 | $\mathbf{0.78 \pm 0.32}$ | $0.55 \pm 0.41$ | $0.66 \pm 0.41$ | $1.00 \pm 0.01$ | $\mathbf{1.00 \pm 0.00}$ | $0.64 \pm 0.17$ |
| Torus-v5 | $\mathbf{0.63 \pm 0.45}$ | $0.29 \pm 0.41$ | $0.43 \pm 0.39$ | $0.67 \pm 0.47$ | $\mathbf{1.00 \pm 0.00}$ | $0.70 \pm 0.08$ |
| Rope-v1 | $0.29 \pm 0.16$ | $0.10 \pm 0.14$ | $\mathbf{0.29 \pm 0.09}$ | $\mathbf{0.66 \pm 0.01}$ | $0.55 \pm 0.06$ | $0.20 \pm 0.08$ |
| Rope-v2 | $0.33 \pm 0.22$ | $0.18 \pm 0.19$ | $\mathbf{0.53 \pm 0.05}$ | $\mathbf{0.71 \pm 0.02}$ | $0.40 \pm 0.18$ | $0.20 \pm 0.01$ |
| Rope-v3 | $\mathbf{0.35 \pm 0.06}$ | $0.19 \pm 0.15$ | $0.30 \pm 0.22$ | $\mathbf{0.52 \pm 0.06}$ | $0.43 \pm 0.21$ | $0.34 \pm 0.06$ |
| Rope-v4 | $0.09 \pm 0.07$ | $0.00 \pm 0.00$ | $\mathbf{0.18 \pm 0.04}$ | $0.37 \pm 0.00$ | $\mathbf{0.42 \pm 0.12}$ | $0.13 \pm 0.01$ |
| Rope-v5 | $0.38 \pm 0.27$ | $0.13 \pm 0.10$ | $\mathbf{0.58 \pm 0.04}$ | $\mathbf{0.70 \pm 0.05}$ | $0.25 \pm 0.11$ | $0.18 \pm 0.03$ |
| Writer-v1 | $0.46 \pm 0.02$ | $0.27 \pm 0.19$ | $\mathbf{0.60 \pm 0.10}$ | $0.78 \pm 0.04$ | $\mathbf{1.00 \pm 0.00}$ | $0.00 \pm 0.00$ |
| Writer-v2 | $\mathbf{0.75 \pm 0.10}$ | $0.14 \pm 0.21$ | $0.30 \pm 0.18$ | $0.86 \pm 0.19$ | $\mathbf{0.98 \pm 0.03}$ | $0.00 \pm 0.00$ |
| Writer-v3 | $\mathbf{0.39 \pm 0.03}$ | $0.34 \pm 0.07$ | $0.36 \pm 0.08$ | $\mathbf{0.81 \pm 0.09}$ | $0.63 \pm 0.27$ | $0.00 \pm 0.00$ |
| Writer-v4 | $0.15 \pm 0.12$ | $0.08 \pm 0.08$ | $\mathbf{0.21 \pm 0.02}$ | $0.28 \pm 0.06$ | $\mathbf{0.49 \pm 0.08}$ | $0.00 \pm 0.00$ |
| Writer-v5 | $0.31 \pm 0.22$ | $0.12 \pm 0.17$ | $\mathbf{0.38 \pm 0.07}$ | $\mathbf{0.35 \pm 0.03}$ | $0.34 \pm 0.06$ | $0.00 \pm 0.00$ |
| Pinch-v1 | $0.00 \pm 0.00$ | $0.00 \pm 0.00$ | $0.01 \pm 0.02$ | $\mathbf{0.16 \pm 0.01}$ | $0.11 \pm 0.04$ | $0.02 \pm 0.03$ |
| Pinch-v2 | $\mathbf{0.16 \pm 0.10}$ | $0.02 \pm 0.03$ | $0.16 \pm 0.13$ | $0.00 \pm 0.00$ | $0.00 \pm 0.00$ | $0.00 \pm 0.00$ |
| Pinch-v3 | $\mathbf{0.06 \pm 0.05}$ | $0.00 \pm 0.00$ | $0.03 \pm 0.04$ | $\mathbf{0.09 \pm 0.02}$ | $0.02 \pm 0.03$ | $0.00 \pm 0.00$ |
| Pinch-v4 | $0.01 \pm 0.02$ | $\mathbf{0.01 \pm 0.03}$ | $0.00 \pm 0.00$ | $\mathbf{0.15 \pm 0.09}$ | $0.00 \pm 0.00$ | $0.00 \pm 0.00$ |
| Pinch-v5 | $0.03 \pm 0.03$ | $0.00 \pm 0.00$ | $\mathbf{0.10 \pm 0.07}$ | $0.02 \pm 0.00$ | $0.00 \pm 0.00$ | $0.00 \pm 0.01$ |
| RollingPin-v1 | $0.47 \pm 0.26$ | $0.10 \pm 0.00$ | $\mathbf{0.83 \pm 0.04}$ | $\mathbf{0.91 \pm 0.04}$ | $0.89 \pm 0.11$ | $0.22 \pm 0.07$ |
| RollingPin-v2 | $0.34 \pm 0.22$ | $0.09 \pm 0.00$ | $\mathbf{0.74 \pm 0.10}$ | $\mathbf{0.92 \pm 0.02}$ | $0.87 \pm 0.04$ | $0.23 \pm 0.09$ |
| RollingPin-v3 | $0.28 \pm 0.34$ | $0.12 \pm 0.00$ | $\mathbf{0.96 \pm 0.05}$ | $\mathbf{0.95 \pm 0.03}$ | $0.82 \pm 0.18$ | $0.36 \pm 0.03$ |
| RollingPin-v4 | $0.42 \pm 0.30$ | $0.15 \pm 0.01$ | $\mathbf{0.87 \pm 0.07}$ | $\mathbf{0.96 \pm 0.03}$ | $0.88 \pm 0.07$ | $0.35 \pm 0.15$ |
| RollingPin-v5 | $0.27 \pm 0.29$ | $0.12 \pm 0.00$ | $\mathbf{0.90 \pm 0.05}$ | $0.91 \pm 0.01$ | $\mathbf{0.98 \pm 0.02}$ | $0.16 \pm 0.08$ |
| Chopsticks-v1 | $0.12 \pm 0.04$ | $0.08 \pm 0.06$ | $\mathbf{0.12 \pm 0.06}$ | $\mathbf{0.92 \pm 0.02}$ | $0.04 \pm 0.05$ | $0.04 \pm 0.03$ |
| Chopsticks-v2 | $\mathbf{0.23 \pm 0.11}$ | $0.22 \pm 0.09$ | $0.22 \pm 0.11$ | $\mathbf{0.98 \pm 0.02}$ | $0.05 \pm 0.05$ | $0.08 \pm 0.11$ |
| Chopsticks-v3 | $0.13 \pm 0.03$ | $0.09 \pm 0.01$ | $\mathbf{0.14 \pm 0.08}$ | $\mathbf{0.88 \pm 0.04}$ | $0.00 \pm 0.00$ | $0.00 \pm 0.00$ |
| Chopsticks-v4 | $0.05 \pm 0.03$ | $0.08 \pm 0.03$ | $\mathbf{0.13 \pm 0.07}$ | $\mathbf{0.81 \pm 0.07}$ | $0.02 \pm 0.02$ | $0.00 \pm 0.00$ |
| Chopsticks-v5 | $\mathbf{0.13 \pm 0.03}$ | $0.09 \pm 0.04$ | $0.10 \pm 0.08$ | $\mathbf{0.81 \pm 0.03}$ | $0.03 \pm 0.03$ | $0.00 \pm 0.00$ |
| Assembly-v1 | $0.00 \pm 0.00$ | $0.00 \pm 0.00$ | $\mathbf{0.00 \pm 0.00}$ | $\mathbf{0.84 \pm 0.03}$ | $0.55 \pm 0.40$ | $0.05 \pm 0.03$ |
| Assembly-v2 | $0.00 \pm 0.00$ | $0.00 \pm 0.00$ | $0.01 \pm 0.00$ | $\mathbf{0.93 \pm 0.02}$ | $0.34 \pm 0.42$ | $0.03 \pm 0.03$ |
| Assembly-v3 | $0.00 \pm 0.00$ | $0.00 \pm 0.00$ | $0.26 \pm 0.31$ | $\mathbf{0.83 \pm 0.18}$ | $0.01 \pm 0.02$ | $0.02 \pm 0.02$ |
| Assembly-v4 | $0.00 \pm 0.01$ | $0.00 \pm 0.00$ | $0.04 \pm 0.06$ | $\mathbf{0.94 \pm 0.02}$ | $0.20 \pm 0.27$ | $0.02 \pm 0.02$ |
| Assembly-v5 | $0.00 \pm 0.00$ | $0.00 \pm 0.00$ | $\mathbf{0.00 \pm 0.00}$ | $\mathbf{0.94 \pm 0.03}$ | $0.24 \pm 0.30$ | $0.01 \pm 0.00$ |
| Table-v1 | $0.01 \pm 0.04$ | $0.05 \pm 0.11$ | $\mathbf{0.57 \pm 0.26}$ | $0.00 \pm 0.00$ | $0.00 \pm 0.00$ | $0.00 \pm 0.00$ |
| Table-v2 | $0.01 \pm 0.02$ | $0.10 \pm 0.14$ | $\mathbf{0.14 \pm 0.17}$ | $0.00 \pm 0.00$ | $0.00 \pm 0.00$ | $0.00 \pm 0.00$ |
| Table-v3 | $0.14 \pm 0.19$ | $0.14 \pm 0.15$ | $\mathbf{0.33 \pm 0.25}$ | $0.02 \pm 0.00$ | $0.00 \pm 0.00$ | $0.02 \pm 0.01$ |
| Table-v4 | $0.05 \pm 0.13$ | $0.19 \pm 0.25$ | $\mathbf{0.34 \pm 0.24}$ | $0.00 \pm 0.00$ | $0.00 \pm 0.00$ | $0.00 \pm 0.00$ |
| Table-v5 | $0.01 \pm 0.01$ | $0.02 \pm 0.04$ | $\mathbf{0.04 \pm 0.03}$ | $0.01 \pm 0.00$ | $0.00 \pm 0.00$ | $0.00 \pm 0.00$ |

Table 6: The normalized incremental IoU scores of each method on each configuration. We run each experiment with 3 random seeds and report their averages and standard deviations (mean $\pm$ std)
.

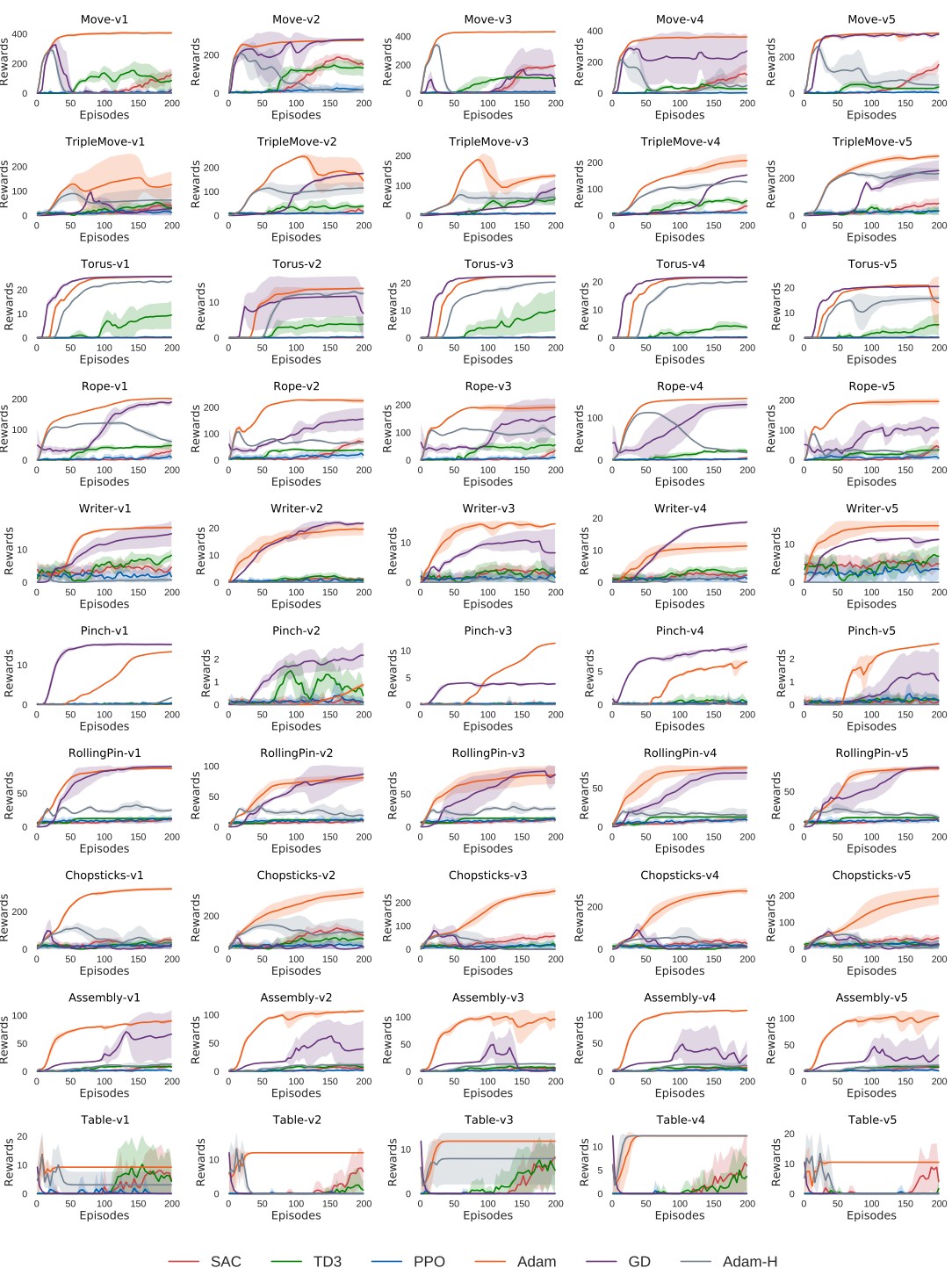

Figure 6: Rewards and variances in each configuration w.r.t. the number of episodes spent on training. We clamp the reward to be greater than 0 for a better illustration.

# F  EXPLANATION OF SOFT IoU

Let $a, b$ be two numbers and can only be $0$ or $1$, then $ab = 1$ if and only if $a = b = 1$; $a+b-ab = 1$ if and only if at least one of $a, b$ is $1$. If the two mass tensors only contain value of $0$ or $1$, $\sum S_1 S_2$ equals the number of grids that have value $1$ in both tensors, i.e., the "Intersection". For the same reason $\sum S_1 + S_2 - S_1 S_2$ counts the grids that $S_1 = 1$ or $S_2 = 1$, the "Union". Thus, the formula $\sum S_1 S_2 / \sum S_1 + S_2 - S_1 S_2$ computes the standard Intersection over Union (IoU). In our case, we assume the normalized mass tensor $S_1$ and $S_2$ are two tensors with positive values. Therefore, we first normalize them so that their values are between $0$ and $1$, then apply the previous formula to compute a "soft IoU," which approximately describes if two 3D shapes match.

