# OpenReview forum: "PlasticineLab: A Soft-Body Manipulation Benchmark with Differentiable Physics"
_ICLR.cc/2021/Conference — ICLR 2021 Spotlight_

### Official Review · AnonReviewer1 · 2020-10-23
**Very interesting and potentially impacting work on soft-body manipulation**

**Rating:** 9
**Confidence:** 3

**Review:**

PlasticineLab

The paper presents a new soft-body manipulation benchmark for RL and differentiable planning.
The presented simulation suite is very interesting and the contribution is solid.

Strength:
- new simulation benchmark with features that are not yet well explored
- differentiable physics to open up possibilities for planning methods
- tasks are difficult enough to be challenging for a while
- baseline results are provided

Weaknesses:
- only the computation times would be good to add

Presentation:
The paper is clearly written and easy to follow.

Ways to improve the paper:
- wall-clock times would really be very useful. Both for the forward pass as well as a backward pass through the entire horizon with Adam. Maybe also some notes on how it can be parallelized since you have a CUDA implementation.


Details:
- p5 last paragraph: "For any grid points with a signed distance d..:" The formulation is not clear enough.
 Do you mean with positive signed distance. Prob not, because you can also have penetration. But why would a point then not have a distance to the rigid body?
- same paragraph: "By definition, s decays exponentially with d until d becomes negative (when penetration occurs)" Well it decays with increasing distance (and then it cannot become negative if it increases...)
- 5.1: IoU definition: Are S always positive? I don't exactly understand what the mass tensor S is. I know it from rigid body dynamics, but this does not seem to be the same here. Can you clarify this better such that it becomes clear why the formula describes and IoU.
- Fig 4. Consider removing the grey background that seaborn uses automatically. The plots will look much cleaner and better visible.

---

> ### Author Response · Authors · 2020-11-22
> **Response to R1**
>
> Thank you for your feedback on our work!
>
> > wall-clock times would really be very useful. Both for the forward pass as well as a backward pass through the entire horizon with Adam. Maybe also some notes on how it can be parallelized since you have a CUDA implementation.
> - We agree that the wall-clock time and details on parallelism are very important information to add, particularly for readers to reproduce our experimental results.
> - As a ballpark number, our simulator’s forward pass currently runs at about 50 fps and is based on the parallelism mechanism discussed in ChainQueen. We plan to add a table that reports detailed wall-clock time in all 10 tasks and a note on parallelism in the appendix.
>
>
> > The formulation is not clear enough; Are S always positive? Consider removing the grey background that seaborn uses automatically.
> - Thank you for pointing out the exposition error in the signed distance! We will clarify the definition of our signed distance function, the IoU definition, and remove the grey background in Fig. 4 in the revised manuscript.

---

### Official Review · AnonReviewer2 · 2020-10-25
**A new simulation benchmark for soft robotics**

**Rating:** 7
**Confidence:** 3

**Review:**

The paper introduces a new open-source simulation benchmark for soft robotics. The simulation environment builds on top of DiffTaichi, an existing differentiable simulator which enables end-to-end differentiability. The paper proposes 10 different tasks, each with 5 variations and evaluates both RL-based policy learning methods and gradient-based optimization methods on those tasks. The results suggests neither current RL-based methods nor gradient-based method can solve most of the tasks efficiently, especially for those require long-term planning.

Overall the paper is well-written and the contribution is well-argued. I have a few comments / questions as follows:
- The simulator only considers the state of the end-effector of the manipulator. It would be great to consider higher DOF soft manipulators (e.g. [1][2]) which would further benefit the soft robotics community.
- Given the randomness and the nature of the RL algorithms [3], the evaluation in section 5.2 should be done with at least multiple random seeds with multiple trials per seed to make the benchmarking results  statistically significant.
- Similarly for Section3, In Table 1, it would be great to see the standard deviation in addition to the average value. It's also not clear how many trials were conducted in order to get the numbers shown in both Table 1 and the plots in Fig. 4.

[1] Della Santina, Cosimo, et al. "Dynamic control of soft robots interacting with the environment." 2018 IEEE International Conference on Soft Robotics (RoboSoft). IEEE, 2018.

[2] George Thuruthel, Thomas, et al. "Control strategies for soft robotic manipulators: A survey." Soft robotics 5.2 (2018): 149-163.

[3] Khimya Khetarpal, Zafarali Ahmed, Andre Cianflone, Riashat Islam, Joelle Pineau. Reproducibility in Machine Learning Workshop, ICML 2018

---

> ### Author Response · Authors · 2020-11-22
> **Response to R2**
>
> Dear Reviewer2, thank you very much for a positive review! We will update the manuscript according to your comments and questions. Below we address the comments one by one.
>
> > The simulator only considers the state of the end-effector of the manipulator. It would be great to consider higher DOF soft manipulators (e.g. [1][2]) which would further benefit the soft robotics community.
> - We are definitely interested in upgrading the kinematic, synthetic manipulators in our submission to more realistic manipulators with higher DoFs, with soft manipulators an ideal candidate. However, as pointed out in the survey paper [2], modeling soft manipulators with actuators in simulation has its distinctive complexity compared to the rigid, kinematic manipulators we have, which we think deserves its own paper and is beyond the scope of this submission.
>
> > Given the randomness and the nature of the RL algorithms [3], the evaluation in section 5.2 should be done with at least multiple random seeds with multiple trials per seed to make the benchmarking results statistically significant.
>
> >Similarly for Section3, In Table 1, it would be great to see the standard deviation in addition to the average value. It's also not clear how many trials were conducted in order to get the numbers shown in both Table 1 and the plots in Fig. 4.
> Multiple random seeds and the standard deviation
>
> - Thanks for the advice! We ran each algorithm with 3 random seeds, and each scene has 5 configurations; therefore, 3x5=15 random seeds in total are used for each algorithm on each scene. We will report the detailed numbers and standard deviations for all experiments.

---

### Official Review · AnonReviewer4 · 2020-10-28
**Authors propose framework consisting of a set of soft-body manipulation tasks. Framework has 10 varied tasks which test different facets of reinforcement-learning algorithms. A differentiable physics engine is used as the core of the framework allowing learning, planning procedures to leverage task gradients while learning to perform each task. Empirical results on multiple state of the art (SOTA) model-free RL models are convincing and show that proposed tasks are too complex for SOTA models.**

**Rating:** 7
**Confidence:** 3

**Review:**

####  Summary

In this work, the authors present PlasticineLab, a new framework for soft-body manipulation tasks for Reinforcement Learning and planning algorithms. The environment consists of a novel soft (i.e., deformable, plastic) material termed Plasticine which is complex to model and manipulate because of the inherently complex high-dimensional governing equations and the large number of degrees of freedom associated with soft materials. The PlasticineLab framework proposes 10 novel tasks involving manipulation of the soft plasticine material. The authors show thorough empirical analysis that traditional state of the art model-free reinforcement learning algorithms fail to effectively learn the task even after a substantial amount of training. Thus effectively showcasing the complexity of the proposed tasks and the inability of state of the art RL models to model the proposed tasks.

#### Positives:

1. Novel tasks: Each task poses a different challenge: E.g., some tasks involve flattening the plasticine, other involve pinching the plasticine while yet other tasks involve grasping one or multiple plasticine objects (at one or multiple points) and deforming it or moving it in some required manner.

2. The variety of tasks test various facets of RL like long-term planning especially in the case of multi stage tasks.

3. Another major effort prelavent in the paper is that the authors have chosen to use a differentiable physics engine using the DiffTaichi system thereby making the gradients available for planning and control algorithms.

4. The paper highlights through empirical results, the superiority of gradient-based approaches (over model-free RL approaches) that leverage the underlying differentiable physics engine toward learning the required tasks.

5. An important facet of a benchmark is to propose tasks that are sufficiently complex for the current state of the art procedures. The authors employ 3 state of the art model-free RL algorithms and show that these RL models perform poorly in a majority of the 10 tasks. Torus, RollingPin, Move tasks are the only three tasks where the model-free procedures are able to perform somewhat comparably with the gradient-based planning approaches which themselves perform well in all but the Writer, Pinch and TripleMove tasks. The last task involves multi-object manipulation and requries long-term planning and hence

#### Concerns:

1. In figure 4. the Adam optimizer and the GD which both seem to consistently accumulate the greatest rewards also have high variance, some commentary about how this can be explained would help the reader better contextualize the results.


2. As one of the main claims of the paper is the challenge of soft-body manipulation and the proposal of a framework for the same, it is imperative to demonstrate the variation of the degree of difficulty with increase (or decrease) in rigidity of the materials being manipulated. A comparative analysis such as this, demonstrating for example the variation in IOU error of the best performing RL model with increasing in yield stress for plasticine would serve to showcase the actual challenge posed by soft-body material mainpulation in the context of the current proposed framework. Ofcourse since decreasing softness and increasing rigidity is most likely not as simple as increasing a single number such as yield stress, this is a minor concern and more a suggestion toward a holistic analysis of the proposed framework.


#### Minor Details & Suggestions:

1. Extrapolation is an important facet of learning algorithms in general. Since one of the suggestions of the current work is to present the PlasticineLab framework as a way to not only characterize RL and gradient-based algorithms but also combine these two families of methods, it is also important to evaluate the performance of these models on unseen but related tasks e.g., manipulating a table with fewer or grater number of legs, trying to place more than 3 objects at specified locations.

2. The citation relating to the paper by Avila et al. titled End-to-end differentiable physics for learning and control published in the Advances in Neural Information Processing Systems conference in 2018 seems to be repeated.

3. Another potential direction of the current framework could be using PlasticineLab to learn policies which might be transferred to the real-world (similar to the task mentioned in [1]). If feasible, adding some brief commentary about this in the context of [1] might open up further avenues of exploration for plasticinelab.

#### References:

1. Matas J, James S, Davison AJ. Sim-to-real reinforcement learning for deformable object manipulation. arXiv preprint arXiv:1806.07851. 2018 Jun 20.

---

> ### Author Response · Authors · 2020-11-22
> **Response to Reviewer4**
>
> Dear Reviewer4, thank you very much for a positive review! We will update the manuscript according to your comments and questions. Below we address the comments one by one.
>
> > In figure 4. the Adam optimizer and the GD which both seem to consistently accumulate the greatest rewards also have high variance, some commentary about how this can be explained would help the reader better contextualize the results.
> - The high variance is due to the fact that rewards from different task configurations (i.e., the same task with parameters slightly perturbed) have varying scales, and we report the aggregated statistics only. We will add tables reporting statistics for each configuration individually.
>
> >  A comparative analysis such as this, demonstrating for example the variation in IOU error of the best performing RL model with increasing in yield stress for plasticine would serve to showcase the actual challenge posed by soft-body material mainpulation in the context of the current proposed framework.
> - Yes, we agree that varying yield stress would be a very interesting experiment. We ran experiments on the Move environment by training SAC with different yield stress. We observed that the agent could achieve a higher reward with an increasing yield stress, demonstrating a correlation between the yield stress and the task's difficulty. More details will be included in the revised manuscript.
> - Additionally, soft-body manipulation tasks' inherent difficulty also comes from the intrinsic high-dimensional state space caused by the elastic deformations. The remaining elasticity still poses many varieties after removing the yield stress.
>
> > It is also important to evaluate the performance of these models on unseen but related tasks
> - Thanks for the suggestion! We strongly agree that evaluating the generalization of the RL algorithm would be a meaningful future direction. Our benchmarks support well on evaluating agents on task variants. However, given RL's current performance in the training environment, we do not expect to achieve good results once the environment changes.
> - Our benchmark supports procedural generation. We believe our benchmark would be a good platform to study the generalization of different algorithms. We will add related discussions in the revised manuscript.
>
> > The citation relating to the paper by Avila et al. titled End-to-end differentiable physics for learning and control published in the Advances in Neural Information Processing Systems conference in 2018 seems to be repeated.
>
> > Another potential direction of the current framework could be using PlasticineLab to learn policies which might be transferred to the real-world (similar to the task mentioned in [1]).
> - We will remove the duplicated reference and add references to sim-to-real papers. We consider sim-to-real problems a very important future direction and will add discussions on our revised manuscript.

---

### Official Review · AnonReviewer3 · 2020-11-04
**Interesting contribution**

**Rating:** 6
**Confidence:** 4

**Review:**


This paper presents PlasticineLab, a differentiable physics environment geared towards softbody manipulation. By implementing a softened rigid-deformable contact interface, and by leveraging recent advances in softbody dynamics simulation (DiffTaichi, ChainQueen), PlasticineLab is able to provide analytical gradients which seem to outperform gradient-estimation based approaches (SAC, TD3, PPO) on few tasks.


## Strengths

**S1** The central problem tackled here is quite interesting (and challenging)! There is a growing interest in the ML community wrt differentiable simulation techniques and in particular, their applicability to learning dynamics.

**S2** The paper is extremely well-written and easy to follow. While this builds heavily on DiffTaichi and ChainQueen, it is commendable that this paper came across as self-contained.

**S3** I believe the characterization of related work is fair. An interesting point made here was that TDW and SAPIEN do not provide assets for soft-body simulation. I am unsure if that's entirely true, but the lack of assets is indeed an issue for softbody simulation. It will be a welcome contribution if this paper were to make these 3D assets publicly available.

**S4** Conceptually, this paper claims that inductive biases (arising from simulation of deformable objects) should be exploited wherever possible. In particular Fig. 4 and Table 1 seem to indicate that gradient-based optimization (using differentiable simulation) consistently outperforms RL techniques in 8/10 tasks. To one's anticipation, gradient-based optimization seems to achieve significantly faster convergence (seems like two orders of magnitude), which is an impressive feat.

**S5** It is interesting to see that it is possible to differentiate through contacts across rigid and deformable objects. To my knowledge, this has not been demonstrated before (has been tangentially discussed in [C]) and is a significant contribution.


## Weaknesses

**W1** The paper could benefit from an explicit exposition of critical design choices that affect differentiability. While PlasticineLab uses a particle-based model for representing and simulating soft-bodies, alternatives in the form of (usually tetrahedral) mesh-based representations exist. It appears that particle systems are chosen to enable trivial differentiability (e.g. the material-point method in the absense of contact forces is analytically differentiable)

**W2** An important detail which I couldn't find in the paper and/or supplementary material is how many of the parameters are simultaneously observable. For example, if the masses of the particles and the manipulator contact parameters are both unspecified, wouldn't this lead to problems in observability (i.e., both quantities cannot be simultaneously solved for, resulting in ill-behaved gradients)?

**W3** Does the approach assume a one-one correspondence between the predicted and target shape? While this might seem a reasonable assumption, I believe gradient computation is cumbersome (and perhaps ambiguous?) were this to be relaxed?

**W4** Another crucial detail that the paper does not seem to get through. As with most other "differentiable physics" approaches, unmodelled effects in the dynamical system might limit the applicability of the system. Since the physics engine only implements forces and softbody dynamics that are "predetermined", I would imagine it is hard to emulate real-world effects such as wear-and-tear, sophisticated contacts, and material properties. In favor of the paper though, I feel this detail might also be out of scope to an extent. (Recent approaches such as Neural dynamical systems [A] and Learning physical constraints by neural projections [B] come to mind, to handle some of these concerns).


## Summary

While the differentiable simulation aspect of the paper is not substantially novel (building atop DiffTaichi, ChainQueen, and soft contact models - Stomakhin et al. 2013), the overall system is impressive and addresses a gap in the differentiable physics community. Simulating differentiable softbody dynamics, as well as interaction with (a limited class of) rigid bodies could open up interesting avenues in reinforcement learning and softbody manipulation.

[A] Neural dynamical systems: balancing structure and flexibility in physical prediction. arXiv 2020

[B] Learning physical constraints with neural projections. arXiv 2020

[C] Scalable differentiable physics for learning and control. ICML 2020

---

> ### Author Response · Authors · 2020-11-13
> **Response to Reviewer3**
>
> Dear Reviewer3, thank you very much for a positive review! Before we address your specific questions on the simulator, we would like to highlight our primary goal is to introduce the first differentiable skill learning benchmark involving elastic and plastic soft bodies. We wish PlasticineLab can lower the barrier of differential physics for machine learning and AI researchers.
>
> We now address individual comments below:
> >The paper could benefit from an explicit exposition of critical design choices that affect differentiability. While PlasticineLab uses a particle-based model for representing and simulating soft-bodies, alternatives in the form of (usually tetrahedral) mesh-based representations exist. It appears that particle systems are chosen to enable trivial differentiability (e.g. the material-point method in the absense of contact forces is analytically differentiable)
>
> - Thanks for the suggestion! More details on this will be included in the revision.  Here we briefly explain why we picked MPM, where material properties are tracked on particles and collisions are handled on the grid:
>     - MPM uses regular grid points for the collision. Grid point collision is much easier to implement and to differentiate than tetrahedra collision. The latter is used in mesh-based approaches such as FEM.
>     - Differentiating through MPM is well-studied and has reliable open-source implementations.  This allows us to focus on extending battle-tested prior work (ChainQueen/DiffTaichi) by adding our own novel features such as differentiable plasticity.
>     - Last but not least, MPM provides both volumetric and particle (point-cloud) representation. Both are very useful for deep learning methods. In our case,  it is very convenient to use sampled particles as inputs to neural networks and calculate the reward based on the volumetric representation.
>
> >An important detail which I couldn't find in the paper and/or supplementary material is how many of the parameters are simultaneously observable. For example, if the masses of the particles and the manipulator contact parameters are both unspecified, wouldn't this lead to problems in observability (i.e., both quantities cannot be simultaneously solved for, resulting in ill-behaved gradients)?
>
> - Our observation only contains the sampled 200 particles' states; We don't take other physical parameters as inputs for training.
>
> > As with most other "differentiable physics" approaches, unmodelled effects in the dynamical system might limit the applicability of the system. Since the physics engine only implements forces and softbody dynamics that are "predetermined", I would imagine it is hard to emulate real-world effects such as wear-and-tear, sophisticated contacts, and material properties. In favor of the paper though, I feel this detail might also be out of scope to an extent. (Recent approaches such as Neural dynamical systems [A] and Learning physical constraints by neural projections [B] come to mind, to handle some of these concerns).
>
> - Thanks for your suggestions. We strive to cover major physical behaviors of plasticine simulation: elasticity, plasticity, and material collision. We strongly agree that real-world effects are a very interesting future direction to explore.
>
> > Does the approach assume a one-one correspondence between the predicted and target shape? While this might seem a reasonable assumption, I believe gradient computation is cumbersome (and perhaps ambiguous?) were this to be relaxed? *
>
> - Sorry, we are not sure if we understand the question correctly. Could you please elaborate more on what "one-one correspondence" refers to?  If you are asking whether we use particle-to-particle correspondences to calculate the gradients, the answer is no. We can define an arbitrary shape as the target in one scene. We don't need the particle correspondence or the target shape's particle distribution to calculate the loss.
>
> Let us know if you have any other questions!

---

### Author Response · Authors · 2020-11-22
**Pre-revision Individual Response Updated**

Dear reviewers,

Thank you very much for your instructive feedback to strengthen this work! We have updated the individual response to each of your reviews under your thread. We will update the revision soon. Please let us know if you have further questions.

---

### Author Response · Authors · 2020-11-23
**General Response: Revision Updated**

We would like to thank the reviewers for their thoughtful feedback. We are glad to see that reviewers generally appreciated the contributions of our paper: the challenging and interesting plastic soft body manipulation task (R1, R3, R4), the use of differentiable physics (R1, R3, R4) and the novelty of our simulation features (R1, R3), the solid experimental results (R1, R4) and the capability to test various facets of RL (R4), the value of our platform to future research (R1, R3), and the writing clarity (R1, R2, R3).

We would like to emphasize again that our main contributions are
- We introduce, to the best of our knowledge, the first skill learning benchmark involving elastic and plastic soft bodies.
- We develop a fully-featured differentiable physical engine, which supports elastic and plastic deformation, soft-rigid material interaction, and a tailored contact model for differentiability.
- The broad task coverage in the benchmark enables a systematic evaluation and analysis of representative RL and gradient-based planning algorithms. We hope such a benchmark can inspire future research to combine differentiable physics with imitation learning and RL.


We have revised our manuscript to include the following changes:
- We have included experiments with multiple random seeds and report both the averages and the standard deviations in Figure 4, and Table 1. (R2)
- We have added discussion on sim2real and generalization in Sec. 6. (R4)
- We have revised the description of the soft contact model in Sec. 4 and the soft IoU in Sec. 5. (R1)

We have also made the following changes in our appendix:
- We have added notes on the parallel mechanism, and Table 2, the wall-clock benchmark, in Sec A (R1).
- We have added Sec. D for the ablation study on the yield stress. (R4)
- We have added Figure 6 and Table 6 in Sec. E to show different approaches' performance for each configuration individually. (R4)

Please don't hesitate to let us know of any additional comments on the manuscript or the changes.

---

### Decision · Program_Chairs · 2021-01-07
**Final Decision**

**Decision:**

Accept (Spotlight)

**Comment:**

This paper proposes a new differentiable physics benchmark for soft-body manipulation. The proposed benchmark is based on the  DiffTaichi system. Several existing reinforcement learning algorithms are evaluated on this benchmark. The paper identify a set of key challenges that are posed by this specific benchmark to RL algorithms. Short horizon tasks are shown to be feasible by optimizing the physics parameters via gradient descent. The reviewers agree that this paper is very well-written, the  problem tackled in it is quite interesting and challenging, and the use of differentiable physics in RL for manipulating soft objects quite intriguing.